# Novel Nucleic Acid Detection for Human Parvovirus B19 Based on *Pyrococcus furiosus* Argonaute Protein

**DOI:** 10.3390/v15030595

**Published:** 2023-02-21

**Authors:** Weiran Chen, Liyang Qiu, Ting Luo, Zhongqin Lu, Xueying Wang, Qi Hong, Jingwen Luo, Lixin Ma, Yuan Wang, Yanming Dong

**Affiliations:** 1State Key Laboratory of Biocatalysis and Enzyme Engineering, Hubei Key Laboratory of Industrial Biotechnology, Hubei Collaborative Innovation Center for Green Transformation of Bio-Resources, School of Life Sciences, Hubei University, Wuhan 430062, China; 2School of Basic Medicine, Hubei University of Arts and Sciences, Xiangyang 441053, China; 3Hubei Jiangxia Laboratory, Wuhan 430200, China

**Keywords:** nucleic acid detection, Parvovirus B19, *Pyrococcus furiosus* Argonaute protein, gDNA

## Abstract

Parvovirus B19 (B19V) is pathogenic to humans and causes various human diseases. However, no antiviral agents or vaccines currently exist for the treatment or prevention of B19V infection. Therefore, developing sensitive and specific methods for B19V infection diagnosis is essential for accurate diagnoses. Previously, a Clustered Regularly Interspaced Palindromic Repeats (CRISPR)-Cas12a (cpf1)-based electrochemical biosensor (E-CRISPR) with a picomole sensitivity for B19V detection was established. Herein, we set up a novel nucleic acid detection system based on *Pyrococcus furiosus* Argonaute (*Pf*Ago)-mediated nucleic acid detection, targeting the nonstructural protein 1 (NS1) region of the B19V viral genome (abbreviated B19-NS1 PAND). Benefiting from independent protospacer adjacent motif (PAM) sequences, *Pf*Ago can recognize their target with guide DNA (gDNA) that is easy to design and synthesize at a low cost. In contrast to E-CRISPR, without preamplification with Polymerase Chain Reaction (PCR), the Minimum Detectable Concentration (MDC) of three guide- or single guide-mediated B19-NS1 PAND was about 4 nM, approximately 6-fold more than E-CRISPR. However, when introducing an amplification step, the MDC can be dramatically decreased to the aM level (54 aM). In addition, the diagnostic results from clinical samples with B19-NS1 PAND revealed 100% consistency with PCR assays and subsequent Sanger sequencing tests, which may assist in molecular testing for clinical diagnosis and epidemiological investigations of B19V.

## 1. Introduction

Parvovirus B19(B19V), which belongs to the *Erythroparvovirus* genus of the *Parvoviridae* family, is a small nonenveloped virus and consists of a linear single-stranded 5.6 kb DNA genome flanked by two identical terminal repeats (ITRs) (Figure 1A) [1]. B19V is pathogenic to humans, and can cause various human diseases, such as fifth disease in children, transient aplastic crisis in patients with chronic hemolytic disorders, non-immune hydrops fetalis, and intrauterine fetal death in pregnant women, as well as chronic pure red blood cell aplasia in immunocompromised individuals [2,3,4,5]. Furthermore, due to the possibility of transmission via blood transfusion and blood products, the contamination of B19V DNA in plasma pools and plasma-derived products can put immunocompromised individuals, hematopoietic-deficient individuals, and pregnant women at high risk of severe complications [5,6]. Consequently, B19V DNA levels must not exceed 10^4^ IU/mL for all plasma-derived products, as was implemented by the U.S. Food and Drug Administration (FDA) in 2004 in order to ensure the safety of these blood products [7]. Though B19V is the etiological agent of many human diseases, there are still no specific antiviral agents or vaccines to treat or prevent B19V infection [8]. Recently, a purine derivative (P7) was discovered which appears to have an antiviral effect, based on a fluorophore-based in vitro nicking assay (FNA) used in the high-throughput screening (HTS) of anti-B19V compounds [9]. Consequently, development sensitive and specific methods for B19V infection diagnosis has dramatically advanced for the sake of accurate diagnoses in B19V epidemiological data collection, monitor viral infections, and support antiviral treatment.

Currently, the diagnosis of B19V infection mainly includes serological and viral nucleic acid tests. Moreover, detecting antibodies in serum remains the cornerstone of B19V infection diagnosis. Immunoglobulin M(IgM) and a >4-fold increase in Immunoglobulin G (IgG) titer or IgG seroconversion in paired serum samples are considered the most valuable markers for B19V acute infection [6]. However, the nucleic acid test is the only diagnostic method for the detection of B19V in immunodeficient individuals. It can also aid in diagnosing acute infections in pregnant women and fetuses with serology tests. For B19V nucleic acid testing, dot blot hybridization, nested PCR/Qualitative PCRs(qPCR), and in situ hybridization are commonly utilized. Among these, nested PCR and qPCR assays are generally used for B19V diagnostic and blood screening in serum/plasma from peripheral blood, other body fluids, fetal cord blood, or amniotic fluid [5,6]. In addition to PCR, in situ hybridization is a valuable complement assay for the detection of B19V DNA in cells and tissue cellular samples, such as bone marrow aspirates, solid tissue biopsies, fetal tissue, or placenta [10]. As an accurate and rapid biosensor, the CRISPR/Cas system provides a potential cost-effective bio-recognition tool for quantifying disease biomarkers, including viral nucleic acid detection [11,12,13,14]. For B19V detection, a CRISPR-Cas12a (cpf1) based E-CRISPR with picomole sensitivity has been successfully established [15]. Nevertheless, given the cost of guide RNA(gRNA) synthesis in the CRISPR/Cas system, the cost-effective *Pf*Ago-mediated nucleic acid detection(PAND) technology was recently developed and applied in the detection of human papillomavirus 16 (HPV-16) and severe acute respiratory syndrome coronavirus 2 (SARS-CoV2) viral nucleic acid in our previous reports [16,17]. As a well-studied prokaryotic Argonaute (pAgo), *Pf*Ago tends to use short 5′-phosphorylated single-stranded DNA (ssDNA) guides to target DNA substrates [18]. During incubating with guide DNA, a molecular beacon, and target DNA PCR product at 95 °C for 20–30 min, *Pf*Ago can bind and cleave the target DNA under the guidance of the guide DNA, generating a short 5′-phosphorylated single-stranded DNA(5′P-gDNA). The generated 5′P-gDNA-bound *Pf*Ago then serves as the guide for a second round of site-directed cleavage to complementary molecular beacons, resulting in the separation of the fluorophore from the quencher. Finally, the fluorescence signal can be detected with a real-time fluorescent qPCR detector or fluorescence spectrometer (as diagrammed in Figure 1A). 

In this report, we have established a unique B19V nucleic acid detection system based on PAND by optimizing gDNA sequences. The diagnostic results from clinic samples with B19-NS1 PAND revealed 100% consistency with PCR assays and subsequent Sanger sequencing tests, which may assist in molecular testing for clinical diagnosis and epidemiological investigations of B19V.

## 2. Methods

### 2.1. Expression and Purification of His-PfAgo Protein from E. coli

The fusion *Pf*Ago protein with a His tag at the N-terminus was expressed in *Escherichia coli* and performed using Ni-NTA affinity purification as described previously [17]. The purified His-*Pf*Ago protein was then analyzed by 8% SDS-PAGE and Western blotting using an anti-His-Tag monoclonal antibody purchased from Proteintech (Wuhan, China).

### 2.2. His-PfAgo Endonuclease Cleavage Activity Assay

The His-*Pf*Ago endonuclease cleavage activity assay was optimized and performed as follows: Briefly, 2 pmol of indicated 5′g-DNA, 5′ phosphorylated by T4 polynucleotide kinases (T4 PNK), and 0.5 pmol of ssDNA (target ssDNA/B19-ssDNA/MB-B19/f-MB-B19) or NS1 DNA PCR product targets were incubated with or without His-*Pf*Ago fusion protein (45 pmol) in a total volume of 20μL *Pf*Ago reaction buffer (20 mM HEPES pH 7.5, 250 mM NaCl, and 0.5 mM MnCl_2_) at 95 °C for 20 min. The cleaved product was then analyzed by 20% TBE-PAGE electrophoresis and followed by SYBR gold nucleic acid dye staining, as previously described [17]. When using molecular beacon f-MB-B19, the fluorescence intensity was measured and analyzed as previously described [17]. For analysis of clinical samples with B19-NS1 PAND, serum blood samples of childbearing-age women were collected from Xiangyang Central Hospital.

### 2.3. The Sensitivity of B19-NS1 PAND Detection

For the sensitive detection of B19-NS1 PAND without PCR, a serial amount of the B19V NS1 target PCR product (achieved with NS1-F/R through PCR amplification) was adjusted to a final concentration of 20 nM, 15 nM, 10 nM, 7.5 nM, 4 nM, 3 nM, 2 nM, or 1 nM and followed by B19-NS1 PAND detection, as mentioned above. For the sensitive detection of B19-NS1 PAND with PCR, an initial concentration of 1.08 × 10^6^ aM (6.52 × 10^7^ copies/μL) B19V infectious clone plasmid (pB19-M20) was determined and followed by a serial of 10-fold dilution to a final concentration from 1.08 × 10^6^ aM to 1.08 aM (6.52 copies). Then, 1 μL of the indicated B19V plasmid was used as a template, and a B19V NS1 target PCR product was achieved with NS1-F/ R through PCR amplification in a total 10 μL reaction volume. Finally, B19-NS1 PAND detection was measured as mentioned above. The details of primers, PCR program, gDNA, and ssDNA targets employed in the B19-NS1 PAND assays are included in Appendix A.

## 3. Results

To establish the B19V nucleic acid detection system based on the *Pf*Ago fusion protein, we first prepared the recombinant His-*Pf*Ago protein described in Methods. As can be seen, the His-*Pf*Ago fusion protein was successfully purified and subsequently confirmed by immunoblot assay (Appendix A). Following the standard reaction assay described previously, the endonuclease activity of the His-*Pf*Ago fusion protein was identified. As shown in Appendix A, the target ssDNA, its phosphorylated gDNA(5′P-gDNA), and the cleavage sites were diagrammed. As expected, a 34 nt cleaved ssDNA strand was produced when *Pf*Ago was incubated with 45 nt positive target ssDNA and its specific 5′P-gDNA (lane 4, Appendix A). Moreover, no cleavage activity was observed without added His-*Pf*Ago or 5′P-gDNA (lane 2,3, Appendix A). These results indicate that the purified His-*Pf*Ago fusion protein does possess endonuclease activity mediated by gDNA, which can be used for subsequent B19V nucleic acid detection.

To guide *Pf*Ago to cleave the B19V DNA targets and the molecular beacon (MB-B19 sstDNA), as diagrammed in Figure 1A, we first designed three gDNAs (gr, gt, and gf) that target the B19V *ns*1 conserved region and one gDNA(gn) that is complementary to the MB. We subsequently performed the endonuclease activity assay, as described above. As expected, when incubated with various 5′P-gDNA (gr, gt, or gf), the His-*Pf*Ago protein could cleave the 46 nt B19V sstDNA (B19 tDNA) to produce 33 nt, 29 nt, and 33 nt ssDNA strands, respectively (Appendix A, lane 2, 4, and 6, Appendix A). Meanwhile, the 5′P-gn can also guide His-*Pf*Ago to cleave the 28 nt MB-B19 sstDNA (MB-B19 tDNA) to produce a 16 nt ssDNA strand (Appendix A, lane2, Appendix A), suggesting that the three designed 5′P-gDNAs can guide His-*Pf*Ago to generate a 5′P-gn product that can cleave the MB. As expected, following the B19-NS1 PAND reaction, the target B19V DNA (B19 NS1-P) was cleaved to generate 5′P-gn, which initiates the second-round cleavage of the fluorescent molecular beacon (f-MB-B19) (Figure 1B, Left panel). Additionally, the replacement of synthesized MB-B19 tDNA with a f-MB-B19 showed that the cleavage activity with *Pf*Ago was 1.5 times higher than the control (Appendix A). These results indicate that the designed gDNA (gr, gt, or gf) and gn can be utilized in B19-NS1 PAND establishment, since the MDC of *Pf*Ago-based or *Thermococcus thioreducens (Ttr*Ago)-mediated detection method ranged from 160 fM to 80 nM [17,19]. To investigate the MDC and sensitivity of the B19-NS1 PAND, a series of diluted NS1 target PCR products were performed according to the procedure. As seen in the middle panel of Figure 1B, the MDC of NS1 target PCR product detection was 4 nM (4 fmol/µL), which was consistent with *Ttr*Ago-based methods (with an MDC of approximately 1 nM) [19]. To achieve even lower detection levels, preamplification with PCR was coupled with B19-NS1 PAND, which is capable of detecting extremely low concentrations of the target, down to ~54 aM (3.26 × 10^4^ copies/mL), much more sensitive than when using *Pf*Ago alone (Figure 1B, right panel). Collectively, these results demonstrate that B19-NS1 PAND combined with PCR has higher sensitivity and specificity, which can be beneficial when it comes to the clinical diagnosis of B19V infection.

Since three gDNAs-mediated PAND increases the reaction’s complexity, we decided to set up one single gDNA-mediated detection system (diagrammed in Appendix A) as demonstrated previously [16]. As can be seen in Appendix A, the 28 nt MB-B19 sstDNA (MB-B19 tDNA) can also be cleaved by *Pf*Ago mediated with gn-2. Meanwhile, as we expected, when introducing a single 5′P-gf, *Pf*Ago cleaved the B19V NS1 PCR target (B19 NS1-p) and generated 5′P-gn-2, resulting in 28 nt fluorescent-labeled molecular beacon(f-MB-B19) being cleaved during the initiation of second-round cleavage (Figure 1C, left panel, lane 2 and lane 4). Replacing the MB-B19 tDNA with the f-MB-B19 also showed that the cleavage activity increased by 50% after adding *Pf*Ago (Appendix A). Finally, as seen in the middle panel of Figure 1C, consistent with the above observation, the MDC of B19-NS1 PAND mediated with single gDNA was also about 4 nM (4 fmol/µL). When introducing the pre-amplification step by PCR assay, similar MDC (54 aM B19V genome) can be detected, as shown in the right panel of Figure 1C. Similarly, these results also indicate that the optimized single gDNA-mediated B19-NS1 PAND has higher sensitivity and specificity than the three gDNAs systems.

Moreover, we performed a B19V DNA test from clinical samples using three guide-mediated B19-NS1 PAND systems. As shown in Figure 1D, the results from B19-NS1 PAND coupled with PCR were in 100% concordance with the PCR test based on the electrophoresis analysis (Figure 1D, left panel), and the subsequent Sanger sequencing test. using six clinical specimens. As B19V DNA levels must not exceed 10^4^ IU/mL for all plasma pools or plasma-derived products, which is lower than 54 aM (5.0 × 10^4^ IU/mL) MDC, we also tried to detect the clinical sample by using the nested PCR assay combined with B19-NS1 PAND. As shown in Appendix A, higher detection sensitivity and specificity were exhibited when nested PCR was introduced, which can be applied to B19V DNA detection in plasma-derived products that must not exceed 10^4^ IU/mL (Appendix A). Finally, similar results were determined when another 32 clinical samples were tested using the nested PCR assay coupled with B19-NS1 PAND (Appendix A).

## 4. Conclusions

Currently, qPCR is the gold-standard technique for most viral nucleic-acid-based clinical diagnostics. In comparison with the qPCR test, programmable site-specific endonucleases from CRISPR- or pAgos-based diagnostic systems have great potential as novel biosensing tools due to their high specificity and efficiency for nucleic acid targets [17,20]. In this report, our B19-NS1 PAND does not require sophisticated laboratory instruments or the expertise of a skilled operator, as only easy-handled standard PCR and simple enzymatic digestion are performed, reducing the cost and complexity of the test [21]. Additionally, the B19-NS1 PAND with no PCR preamplification can shorten the detection time from greater than one hour to just 30 min per batch. In comparison, the single-nucleotide specificity of *Pf*Ago allows multiple nucleic acid targets to be detected by PAND within one assay [16,17]. Once the selected gDNAs that distinguish different B19V genotypes are available, PAND can also be employed to detect multiple B19V genotypes, similar to HPV or SARS-CoV-2 detection assays [16,17]. In addition, compared with CRISPR-based detection systems, PAND exhibits unique features for the detection of multiple targets. Currently, it is still a challenge to establish more than four channel multiplex detection based on the CRISPR/Cas system due to the number limitation of different CRISPR effectors [20,22]. However, due to the specific cleavage of molecular beacons in PAND, which is dependent on base pairing between gn and its specific molecular beacons, it has been demonstrated that PAND can detect five HPV subtypes in one reaction using five different fluorophores [17]. Therefore, the advantage of PAND for multiple nucleic acid detection is far greater than that of CRISPR-based systems. Moreover, the independent PAM for B19-NS1 PAND not only provides more choices in guide selection, but also enables the positioning of the gn to be optimized for the highest sensitivity of detection. Additionally, replacing the gRNA with short stable single-stranded DNA as a guide can also reduce the cost.

Of course, this method also has some pending problems. For qPCR, absolute or relative quantification can be performed; however, absolute quantification for CRISPR-based diagnostics can only be achieved within the picomolar-to-micromolar range [23]. Furthermore, for clinical samples that require a lower MDC and thus need preamplification, quantification with a CRISPR-based detection system cannot easily be achieved, as the saturation during preamplification can prevent a quantitative readout [23]. Our results have also shown that only approximate linearity between the *Pf*Ago-based collateral-cleavage activity and its target concentration can be obtained without preamplification. Given the limitation of this technique, it is preferable for qualitative detection rather than the quantitative evaluation of B19V samples. Moreover, the challenge of obtaining higher sensitivity without PCR and achieving a one-tube reaction needs to be addressed in the future.

Herein, we have established a novel nucleic acid detection system based on the *Pf*Ago-mediated targeting of the B19V genome. Without preamplification with PCR, three guide- or single gDNA-mediated B19-NS1 PAND have a MDC of 4 nM, which is approximately 6-fold higher than that of E-CRISPR. When introducing a standard PCR step for B19V target amplification, the MDC can be dramatically increased to 54 aM, consistent with previous *Pf*Ago- or *Ttr*Ago-based methods [17,19]. In addition, the platform was validated to detect B19V DNA in clinical samples with high specificity and sensitivity, providing essential tools for the detection of other parvoviruses. Unfortunately, since the B19V DNA level in plasma-derived products should not exceed 10^4^ IU/mL, the 54 aM MDC is too high for B19V DNA detection in plasma pools when its viral concentration is low, meaning that further improvements are necessary for clinical applications.

## Figures and Tables

**Figure 1 viruses-15-00595-f001:**
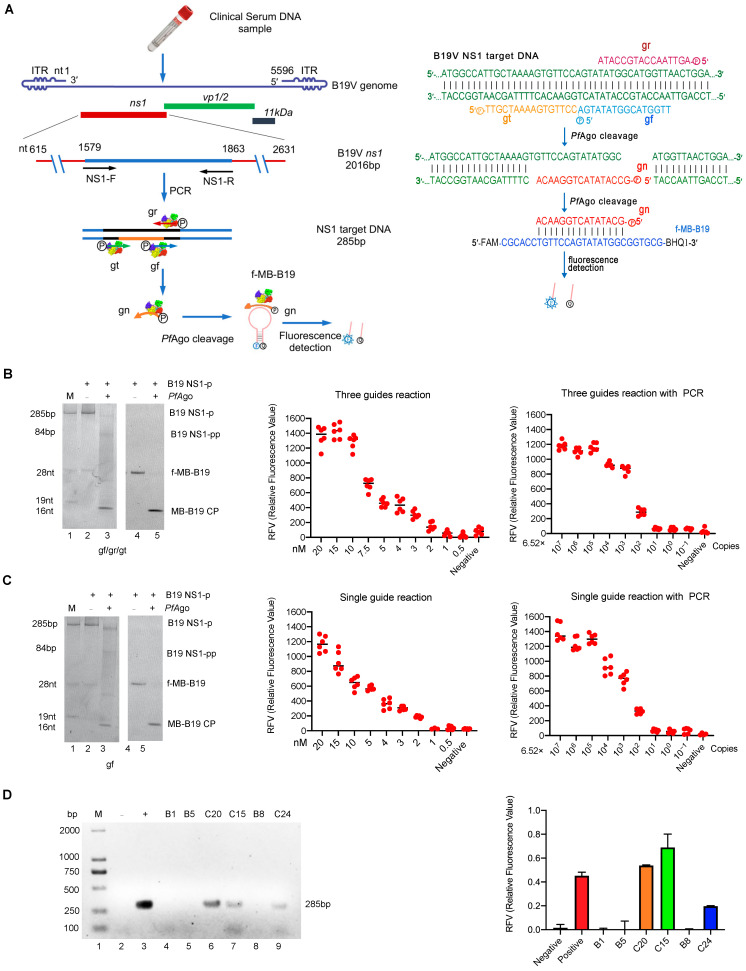
Establishment of the B19-NS1 PAND and its application in the clinical sample. (**A**) Schematic of B19-NS1 PAND workflow. The nucleotide location of the NS1 target region in B19V genome, the size of the PCR product, and the cleavage bands are indicated. The NS1 target PCR product, PfAgo cleavage region and the newly generated gn are indicated in blue, black and grown color line respectively. The three 5′-phosphorylated single-stranded DNA guides (gr, gt, and gf) are indicated in red, green, and blue arrow respectively. (**left panel**). The sequences of three 5′-phosphorylated single-stranded DNA guides (gr, gt, and gf), newly generated single-stranded DNA (gn), and molecular beacons are shown and highlighted in different colors (**right panel**). f-MB-B19, molecular beacon; gn, newly generated single-stranded DNA guide; gr, gt, and gf, three 5′-phosphorylated single-stranded DNA guides of B19V; Q, quencher; F, fluorophore. Establishment of B19-NS1 PAND system with three guides (**B**) or one guide (**C**). TBE-PAGE analysis of B19-NS1 PAND system with three guides (**B**, **left panel**) or one guide (**C**, **left panel**) was stained with SYBR Gold dye (Lane 1–3) or the fluorescence image was recorded directly by UV (Lane 4–5). The sizes of the target, gDNA, molecular beacon, and cleavage product bands are indicated in the left panel. The abbreviations for these components are gn/gn-2 target ssDNA; MB-B19 CP, cleavage product of molecular beacon; B19 NS-p, B19V NS1 PCR product; B19 NS-pp, B19V NS1 PCR product cleavage product. The MDC analysis of the three gDNA mediated B19-NS1 PAND without PCR (**B**, **middle panel**) or with PCR (**B**, **right panel**); MDC analysis of the one gDNA-mediated B19-NS1 PAND without PCR (**C**, **middle panel**) or with PCR (**C**, **right panel**). Relative fluorescence intensity detection at a series of diluted NS1 target PCR products is shown. (**D**) Analysis of clinical samples with B19-NS1 PAND was conducted. Six clinical samples (C15, C20, C24, B1, B5, B8) detected by standard PCR assay were analyzed with agarose gel electrophoresis using ethidium bromide staining (**left panel**), or were coupled with three gDNAs-mediated NS1 PAND systems (**right panel**). Negative and positive controls are indicated with minus and plus. These data were collected based on three independent experiments and are presented as the mean ± SD.

## Data Availability

Not applicable.

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
