# Peer review of "Novel Nucleic Acid Detection for Human Parvovirus B19 Based on Pyrococcus furiosus Argonaute Protein"

_viruses, 2023, doi:10.3390/v15030595_

Round 1

Reviewer 1 Report

In this paper entitled "Novel nucleic acid detection for Human Parvovirus B19 based on Pyricoccus furiosus Argonaute protein", the authors propose a new method for the detection of B19 genome using Pyrococcus furiosus Argonaute protein. Taking advantage of its endonuclease properties, they used combinations of a set of primers and guides designed to recognize NS1 gene to drive its cleavage. Cleaved product were then quantified, leading to detection of B19 genome, from either plasmid or clinical samples. Data are organized in a main figure with 4 panels (A to D), and 4 supplementary data (one table and 3 figures). The entire paper is sustain by 18 bibliographic references, 15 of them illustrating the introduction and 3 references in conclusions chapter.

Overall, this paper presents a promising technique to analyze B19 genome with improved sensitivity compared to previously described techniques (conventional PCR and E-CRISPR (electrochemical biosensor based detection). However conclusions are overstated, promising an increase of sensitivity without performing side by side experiments using concurrent techniques on same samples. B19 detection should be carried out in parallel using conventional techniques, such as PCR or by the E-CRISPR (electrochemical biosensor based detection) technique described by the authors.

Presented experiments and data are based on a low number of replicates per point (n=3), with no statistical analysis to support conclusions. As for example, the main paper figure proposes 4 graphic representation on panels B and C. Some conditions show a large dispersion between measurements, especially between 20 to 5 nM. The number of replicates should be increased to support increased sensitivity.

Moreover, a detection technique has to be reproducible to sustain proper detection, and define maximal and minimal range of detection, as well as proportionality zone between template substrate (here B19 genome) and method of detection. Results as shown in figure 1 failed to define such linearity, essential for quantitative detection of B19 genome: in figure 1 B, detection seems to give same results between 20 to 7.5 nM (107 to 104 copies), to drop then to quite basal level below 5 nM (or 103 copies with PCR). As no differences are seen for concentration higher or below 104 copies, results raised a limitation to this technique, classifying it to qualitative detection rather than quantitative evaluation of B19 samples. Authors should extensively discuss this point in conclusion chapter. To illustrate this limitation, B19 copie numbers in clinical samples should be indicated to properly appreciate the gain of using this method, knowing that the main interest would be to favor false-negative samples (while B19 positive) in PCR to challenge them with this new technique.

Minor comments:

1) The main figure is too complex and dense: the schematic illustration (1A) is small and weakly explained with no legend included to understand.  The representation of B19 ITR is unconventional, showing 2 different ITR structures while 5’ and 3’ B19 ITR share identical structures (please see in schematic representation in cited reviews or ICTV report).

2) Introduction is based on 4 review articles among references to illustrate B19 literature. Please use more article references.

3) In figure 1B and C, please use the same X and Y-axis to facilitate reading of the figures.

4) Abbreviations should be clearly indicated and explained at their first appearance in text. For example, see lines 20, 22, 23 in abstract, lines 66, 72 in introduction.

5) Introduction could be clarified by clearly explaining existing B19 detection techniques, and differentiating them according to their field and use (clinical, patient bedside versus plasma-derived industry and HTS development).

6) Some sentences are unclear: line 39-42, line 89 (please clarify);

7) Line 88: please replace by proper reference (13)

8) References cited together (lines 58 and 64) should be chronologically numbered according to the same rules.

Reviewer 2 Report

The authors of the brief report “Novel nucleic acid detection for Human Parvovirus B19 based on Pyrococcus furiosus Argonaute protein” report on nucleic acid detection system based on the PfAgo-protein targeting B19V genome.

This follows papers published in 2019 and 2021 on PfAgo-mediated nucleic acid detection technology for HPV-16 and SARS-CoV-2 nucleic acid detection. The reported paper used the same method for development B19V detection system. Considering B19V topicality, new, sensitive and specific cost-effective and rapid tests for B19V infection detection is highly requisite.

General concept comments:

In general, the presented research is relevant, interesting and adequate and the paper is topical for the field.

Major revision is necessary:

Revise ethical statements (including informed consent), considering manuscript includes human clinical samples.

It is necessary to draw conclusions and rewrite the existing section. The conclusions must be consistent with the evidence of this work and arguments presented. Do not repeat abstract, introduction or results within conclusions.

The citations preferably to first published source instead of recent review.

Keep consequence with the abbreviations throughout the text – first full name, then abbreviation only.

Please explain what it refers to: line 42 – “…reduce the B19V DNA contamination”.

Integrate materials and methods subsection “Oligos, enzymes, and reagents” within methods.

Review the use of caps locks and add/remove spaces throughout the manuscript.

Lane 88-89: reference must be included in the list or reference number (13?) included in the text.

No need to repeat previously published method if reference is added.

The methods can be shorter – less like protocol. Details include in supplements.

Lane 150: “1.5 times than” missing word.

Figure S2C add bar labels.

Split figure 1 into 4 figures and left-right panels label with A, B, C. Legend must be shortened, excluding method details.

AIs sample size 6 adequate for generation of reliable results and conclusions?

Please describe advantage of the detection system over existing ones, including quantitative real time PCR.

Round 2

Reviewer 1 Report

Authors should increase the number of replicates in order to sustain properly conclusions. Last chapter of the manuscript (conclusion) should provide a detailed comparaison between available detection aproaches and present a consistent conclusion to clearly position this new detection approach among existing ones. 

Information of patients should be include to fullfill ethical concerns. English et typography could be improved. 

Regarding response to reviewers on first round, please answer properly and point-by-point to the questions raised by each reviewer separately rather than presenting a unique compilation for both reviewers. It is disturbing and hardly readible.

Reviewer 2 Report

Informed consent for the surveillance study of B19V DNA 476 detection must be included.

Please provide manuscript with conclusions. The current text corresponds to discussion.

There are still abbreviations in text and supplementary that must be corrected.
